# Piston Sensing for Golay-6 Sparse Aperture System with Double-Defocused Sharpness Metrics via ResNet-34

**DOI:** 10.3390/s22239484

**Published:** 2022-12-04

**Authors:** Senmiao Wang, Quanying Wu, Junliu Fan, Baohua Chen, Xiaoyi Chen, Lei Chen, Donghui Shen, Lidong Yin

**Affiliations:** 1Jiangsu Key Laboratory of Micro and Nano Heat Fluid Flow Technology and Energy Application, School of Physical Science and Technology, Suzhou University of Science and Technology, Suzhou 215009, China; 2Soochow Mason Optics Co., Ltd. of Graduate Workstation in Jiangsu Province, Suzhou 215028, China; 3Suzhou Dechuang Measurement & Control Technology Co., Ltd. of Graduate Workstation in Jiangsu Province, Suzhou 215128, China; 4Zhangjiagang Optical Instrument Co., Ltd. of Graduate Workstation in Jiangsu Province, Suzhou 215006, China; 5Currently School of Electronic and Optical Engineering, Nanjing University of Science and Technology, Nanjing 210094, China; 6Suzhou Mason Optical Co., Ltd., Suzhou 215028, China

**Keywords:** Golay-6, sparse aperture, piston sensing, ResNet-34

## Abstract

In pursuit of high imaging quality, optical sparse aperture systems must correct piston errors quickly within a small range. In this paper, we modified the existing deep-learning piston detection method for the Golay-6 array, by using a more powerful single convolutional neural network based on ResNet-34 for feature extraction; another fully connected layer was added, on the basis of this network, to obtain the best results. The Double-defocused Sharpness Metric (DSM) was selected first, as a feature vector to enhance the model performance; the average RMSE of the five sub-apertures for valid detection in our study was only 0.015λ (9 nm). This modified method has higher detecting precision, and requires fewer training datasets with less training time. Compared to the conventional approach, this technique is more suitable for the piston sensing of complex configurations.

## 1. Introduction

Deep-space exploration makes significant demands on the imaging abilities of telescopes [1]: thus, the optical sparse aperture (OSA) becomes a substitute mode for high-resolution imaging, being like a large monolithic aperture with lower size and lighter weight [2]. The Golay-6 configuration is a classic non-redundant sparse aperture array, first proposed by Marcel J.E. GOLAY [3]. In addition, Boeing-SVS Inc has completed a Small Business Innovative Research (SBIR) project called low-cost space imager (LCSI), that is based on the Golay-6 OSA system [4]: its significance is to control the deviation among the sub-apertures known as pistons to within a fraction of a wavelength [5]; therefore, piston sensing is needed, to gain better optical performance.

Many methods have been used for the piston error detection of the OSA system, including the Schack Hartmann sensor, which can detect the piston error between the sub-mirrors of the micro-lens array [6]; however, the low sensing accuracy of the Schack Hartmann sensor only satisfies coarse phasing with a limited detection range, and suffers 2π ambiguity. Another mature approach, Phase Diversity (PD), extracts the piston through a pair of focused and defocused images: it breaks the limits of the imaging content, and could be applied to the expanded target [7]; however, Phase Diversity (PD) is cumbersome, so the calculations waste much time, and the 2π ambiguity also affects this method.

In the past few years, deep learning has produced an outstanding achievement: researchers have proposed an advanced convolutional neural network (CNN) that can directly estimate the Zernike coefficients of an incoming wavefront from an intensity image, and use this for piston sensing [8]. Guerra-Ramos et al. used two shallow CNNs to learn the intercept of the feature model’s ambiguity range and piston step values [9]. Xiafei Ma et al. demonstrated that using only a single deep convolutional neural network (DCNN) is sufficient to detect pistons from a broadband extended image [10]. Yirui Wang et al. authenticated that a Bi-GRU neural work with a much simpler structure can be effectively used for delicate phase segmented mirrors [11].

In this paper, we applied a deep network to the Golay-6 sparse aperture’s piston sensing. By comparison, we demonstrated that a single ResNet-34 neural network can effectively detect pistons with high precision and better training efficiency from feature images. We also used focused and defocused images to build feature vectors, and chose the DSM as a higher quality training sample, to improve the sensing accuracy and precision. In addition, we added a fully connected layer, to further enhance ResNet-34’s performance. Our proposed method realizes high-precision piston sensing for a complex configuration OSA system.

## 2. Method

### 2.1. Basic Optical Principle and Alternative Metrics

The Golay-6 OSA system consists of six sub mirrors of the same size, which form a large aperture. According to the principle of diffraction-limited incoherent imaging, the image captured on the focal plane of the Golay-6 system can be modelled as
(1)$$S=(FT\{|FT[P(u,\lambda )]{|}^{2}\})\cdot [FT(O)]$$
where P(u,λ) is the generalized pupil function expressed by a 2D vector u, λ is the wavelength, FT(·) represents Fourier transform, O is the object, and S is the Fourier transforms of the focused image, respectively. When piston errors are included in the system, the generalized pupil function changes as follows:(2)$$P(u,\lambda )=p(u-u_{1})+\sum_{2}^{N}\left[p(u-u_{n})\cdot \exp(\frac{2\pi i}{\lambda }\cdot \phi_{n})\right]$$
where p is the pupil function of the sub-aperture, N is the total number of all the pupils, equal to 6 in this case, $u_{n}$ represents the central position vector of the corresponding pupils, and $\phi_{n}$ represents the piston error of each aperture. When including the defocus aberration in the generalized pupil function, Equation (2) will become
(3)$$P_{defocus}(u,\lambda )=P(u,\lambda )\cdot \exp[\frac{2\pi i}{\lambda }\cdot \Delta \psi_{k}(u,f,D)]$$
where A is the binary pupil function of the minimum surrounding circle that encloses all the pupils, $△\psi_{k}$ represents the defocus aberration, f is the focal length, and D is the diameter of the surrounding circle.

We chose several feature vectors for network learning using focus diversity images, also known as Gonsalves Metrics [12], which can be expressed as
(4)$$E=\sum_{}\frac{{\left|S_{o}{\overset{\wedge }{H}}_{d}-S_{d}{\overset{\wedge }{H}}_{o}\right|}^{2}}{{\left|{\overset{\wedge }{H}}_{o}\right|}^{2}+{\left|{\overset{\wedge }{H}}_{d}\right|}^{2}}$$
where $S_{o}$ is the Fourier transform of the focused images, $S_{d}$ is the Fourier transform of the defocused images, $H_{o}$ represents the focused optical transfer function (OTF), and $H_{d}$ represents the defocused OTF. The $\overset{\wedge }{}$ implies the estimate operator [12]. If two defocused images are used to construct the DSM, Equation (4) can be written as
(5)$$M_{double-sharpness}=\frac{\left(S_{o}\cdot S_{d1}^{*}-S_{o}^{*}\cdot S_{d1})+(S_{o}\cdot S_{d2}^{*}-S_{o}^{*}\cdot S_{d2})+(S_{d1}\cdot S_{d2}^{*}-S_{d1}^{*}\cdot S_{d2}\right)}{S_{o}\cdot S_{o}^{*}+S_{d1}\cdot S_{d1}^{*}+S_{d2}\cdot S_{d2}^{*}}$$
where $S_{o}$ is the Fourier transform of the focused images, $S_{o}^{*}$ is its complex conjugate, $S_{d1}$ and $S_{d2}$ are the Fourier transforms of the two different defocused images, and $S_{d1}^{*}$ and $S_{d2}^{*}$ are their complex conjugates. The focused and defocused images can be obtained according to Equation (1).

### 2.2. ResNet-34 Structure and Loss Function

As described in the first part, CNN’s shallow layers limit the feature extraction, and make it impossible to identify all pistons from the feature vector. We used a ResNet-34 neural network to solve these problems, the added layers of which ensure extracting more complex image features, and need fewer training data. To obtain the best detection results, a fully connected layer was added to the ResNet-34 network, to enhance the network’s nonlinear expression ability, and to further improve our method’s precision, as shown in Section 3.3.

The architecture of ResNet-34 is shown in Table 1 [13]. Unusually, ResNet-34 has four groups of Basic Block, each comprising several residual modules that avoid the gradient explosion problem, and ensure the performance will not decline. Batch normalization accelerates training, and ReLU activation functions realize the feature extraction.

The loss function of the piston sensing is defined as follows:(6)$$L_{MSE}=\frac{1}{n}\sum_{i=1}^{n}{\left(y_{predi}-y_{labeli}\right)}^{2}$$
where $y_{predi}$ is the prediction and output of the network, $y_{labeli}$ is the ground truth as label, in company with the input feature vectors, and n is the number of the sub-apertures. By iteratively calculating loss, the predicted value will be closer to the sample truth value, making the prediction more accurate.

### 2.3. Data Sets and Training

Like the previous studies [9,10], we also used simulated data to validate our method. First, we built the Golay-6 OSA system, as shown in Figure 1. The diameter of the surrounding circle approximated 24.17 mm, and the diameter of the sub-aperture was 5 mm, with a filling factor roughly equal to 25.68%. The system’s focal length was 1000 mm.

After setting up the whole system, we used a resolution target as the object, to gain the focused image and its defocused counterpart, illuminated by the light of 600 nm, as shown in Figure 1. We built three data sets based on different metrics, as shown in Figure 2a–c: each had 9000 images for the training and 1000 for the testing. Every image in a group was loaded with different pistons, and the values of the corresponding picture in both groups were the same. As the piston error is periodic in the imaging performance for the OSA system [14], the values ranged from 0 to λ, and the pistons ranged from 0 nm to 600 nm. We set sub-aperture 1 as the reference, and randomly generated pistons among the remaining five sub-apertures.

The whole process is shown in Figure 3, including training and testing procedures, where Sub-a0 represents the sub-aperture 0. In training, we used DSMs as the feature vectors for the inputs, and the corresponding pistons as labels were also input as the ground truth, while using the Power Metrics (PM) and the Sharpness Metrics (SM) as a contrast to explore the detection precision and accuracy. We used two defocused images and one focused image to build the DSM. The defocused images had the same defocus distance but opposite directions.

To ensure independence, we adopted the PM as the network input in Section 3.1. PM images have fewer feathers, which can better test the actual extraction ability and learning ability of different networks.

In testing, we first imaged a Golay-6 OSA system with unknown pistons, to obtain a set of focused and defocused maps for constructing the test feature vector. Then, we put it into the well-trained network. Finally, it could quickly predict all pistons for the five sub-apertures.

When the training data set was ready, we built the ResNet-34 in Python for piston sensing. The proposed network was run and trained on a GPU (NVIDIA GeForce RTX3060 laptop GPU). The experimental environment is presented in Table 2.

## 3. Results

### 3.1. Performance and Comparison between ResNet-34, VGG-16, and Alex Net

According to the recorded experimental data, ResNet-34 learned from 9000 feature vectors based on Power Metric with 100 epochs in about 172 min. The estimation for a single image took only about 88 ms, which realized efficient testing. The training time of the VGG-16 network reaches 280 min. Evidently, ResNet-34 is faster. Figure 4 shows the respective loss function curves over the whole training section:

The graphs show that all these networks converged after learning 100 times against the same training data set. In comparison, the training loss and test loss of the ResNet-34 had astringency, and performed better than Alex Net; however, VGG-16 was slightly inferior, reflecting that its loss had not been significantly reduced; by inference, ResNet-34 may have higher precision.

Considering that sensing accuracy is an important criterion, we used the predicted piston values minus the ground truth values, and took the absolute values as the error; the values beyond 60 nm (0.1λ) were regarded as invalid. The sensing accuracy represented the ratio of the number of valid predictions to the total number of predictions, which could be calculated by counting the number of errors within the certain range in the test.

To test the precision of our proposed method, we separately calculated the RMSE between the valid predicted pistons with their corresponding ground truth value for all the sub-apertures, and took its average value as the final evaluation. Taking sub-aperture 0 as an example, Figure 4 shows the error distributions of the test sets for three networks. We also give details of all five sub-apertures in Table 3.

As shown in Figure 5a–c, most of the errors of ResNet-34 were located within the region of [0, 15] nm, while those of Alex Net were in the range of [0, 60] nm, but most of the results of VGG-16 were greater than 60 nm. In Table 3, we can see that the average sensing accuracy for all the sub-apertures of ResNet-34 reached 93.78%. Furthermore, the ResNet-34’s average RMSE was 12.58 nm—18.36% lower than that of Alex Net, and 59.41% lower than VGG-16. The sufficiently small detection error indicates that ResNet-34 achieves high-precision piston sensing.

### 3.2. Performance of the Piston Sensing, Based on Double-Defocused Sharpness Metrics

While PM performed well on ResNet-34, there was room for improvement in sub-aperture 5’s detection. To further improve the precision and accuracy of our proposed method, we used the DSM as input for training and testing on ResNet-34.

The scatter diagram of the corresponding testing results of sub-aperture 5 is displayed in Figure 6a–c. We can see that both the DSM and the SM samples converged well within a specific range, and that the DSM’s error was more centrally distributed. With few errors ignored, the two Sharpness Metric splash ranges were more concentrated than the PM: this shows that the DSM has higher precision.

We also displayed the sensing results comparison of all the five sub-apertures based on the DSM, the SM, and the PM in Table 4. Compared to the others, the DSM’s average sensing accuracy went up to 96.8%, guaranteeing the correctness of its detection; meanwhile, its average RMSE reduced to 9.74 nm. The high precision ensured that the error of the five sub-apertures was less than 0.027λ (16 nm), which sufficed for the fine phasing.

### 3.3. Further Improvements

We continued to improve the proposed method, to obtain better detection performance. Our initial ResNet-34 network did not use a softmax classifier, but rather a fully connected layer, to directly gain the distributed feature representation, which was the predicted value: on this basis, we added another fully connected layer to reduce the number of parameters. We used the modified network to learn and test again; the results are shown in Table 5. The new network improved the parameters for the same data sets based on the three metrics: specifically, the DSM’s average sensing accuracy went up to 97.26%, and its average RMSE reduced to 9 nm. Compared with other research [10], which used 120,000 images to build the training data set for the piston sensing of the Golay-6-1 systems, and whose detection error was about 24 nm, our detection was more accurate, and our method relied less on the data sets, which made it more feasible.

## 4. Discussion

We used Local Binary Pattern (LBP) to analyze the reasons for the different performances among the DSM and others, and explored their specific features loaded with the same piston errors [15]. By contrast, we could see fewer features in Figure 7a. As a result, the number of features extracted by the network based on the PM was limited, and it was more challenging to recognize. The local feature tracing in the yellow outline in Figure 7c shows that the basal part of the SM and the DSM was abundant. In addition, the distinction between the features in the yellow outline is conspicuous.

Therefore, this explains how the SM and the DSM can learn more rich features. Compared with the SM in Figure 7b, the same area in the red outline of the DSM has a high gray value, and the scale is shallower for non-primary features. In addition, the main features in the yellow outline of the DSM are more emergent than the SM; thus, we may indicate that the DSM can avoid learning redundant features, to guarantee better precision prediction, which fits well with Yan’s research [16].

For actual implementations, further imaging experimentation is still needed in future developments. As for the role of the network, further work is needed, to consider reducing the training time and cutting the number of data sets. In addition, an advanced computing device could also accelerate computation speed.

## 5. Conclusions

This paper improves the existing deep-learning piston detection method with a more advanced ResNet-34 network. By simulation, we verified the feasibility and efficiency of the proposed method. Specifically, the average detection RMSE achieved 9 nm, and the sensing accuracy reached 97.26%: this proves that our method, based on the Double-defocused Sharpness Metric (DSM), breaks the limits of the structural redundancy on sensing accuracy for a complex non-centrosymmetric array, such as a Golay-6 system. By using the Uniform Pattern LBP to extract features from different metrics with the same pistons, we conclude that the DSM could relieve the impact of feature redundancy on network performance, and improve sensing accuracy. Based on the above advantages, our proposed method could be widely applied to phasing the OSA telescope.

However, this method needs to collect a large number of real images, and to preprocess the corresponding piston values, which requires a significant amount of time for preliminary preparation.

## Figures and Tables

**Figure 1 sensors-22-09484-f001:**
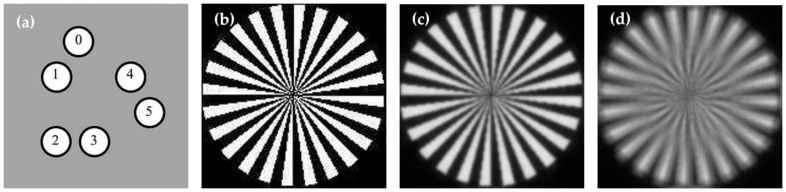
Golay-6 structure and its simulated imaging results: (**a**) configurations of Golay-6 arrays; (**b**) imaging target; (**c**) the simulated imaging map of Golay-6 system (focused image); (**d**) the defocused imaging map.

**Figure 2 sensors-22-09484-f002:**
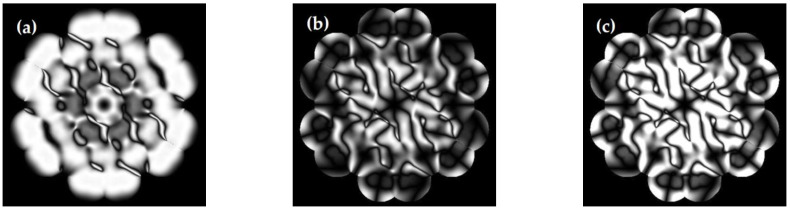
Examples of the training data for the Golay-6 system, based on the Alternative Metrics carried by the same pistons: (**a**) the Power Metric (PM); (**b**) the Sharpness Metric (SM); (**c**) the Double-defocused Sharpness Metric (DSM).

**Figure 3 sensors-22-09484-f003:**
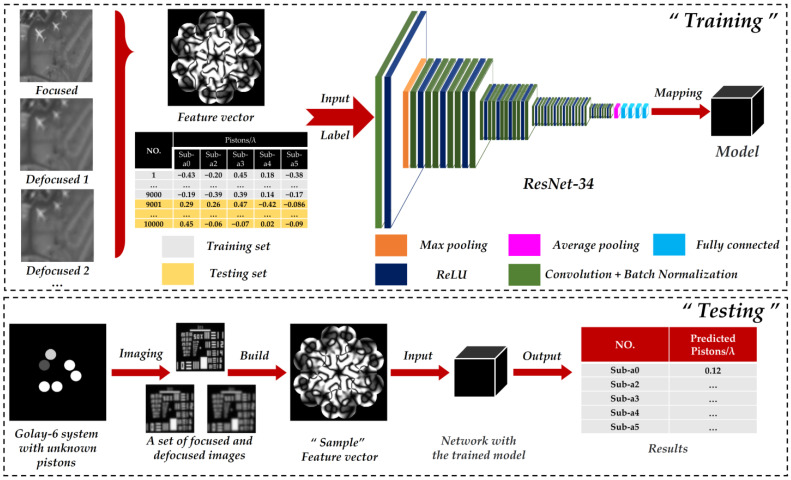
The principle of the piston sensing method based on ResNet-34, including the detailed process of “Training” and “Testing” procedures.

**Figure 4 sensors-22-09484-f004:**
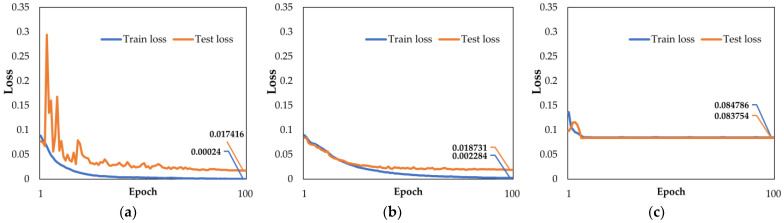
The evolution of the loss function during training for the Golay-6 OSA system based on Power Metric: (**a**) ResNet-34; (**b**) Alex Net; (**c**) VGG-16.

**Figure 5 sensors-22-09484-f005:**
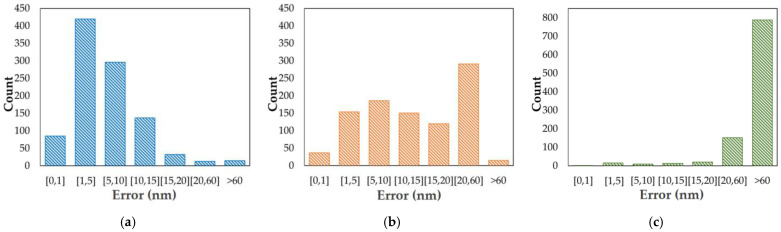
The error distributions of 1000 test feature vectors, using sub-aperture 0 as an example: (**a**) the results of ResNet-34; (**b**) the results of Alex Net; (**c**) the results of VGG-16.

**Figure 6 sensors-22-09484-f006:**
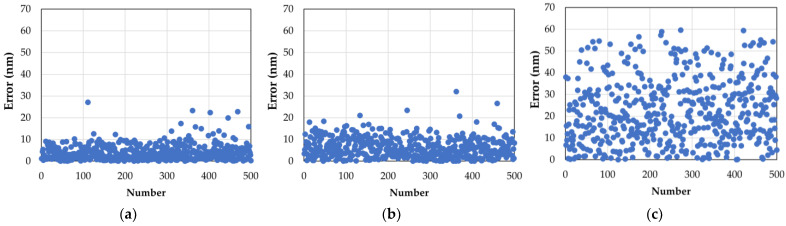
The testing results of the different data sets: (**a**) 500 errors were selected from the testing data set based on the DSM; (**b**) 500 errors were selected from the testing data set based on the SM; (**c**) 500 errors were selected from the testing data set based on the PM.

**Figure 7 sensors-22-09484-f007:**
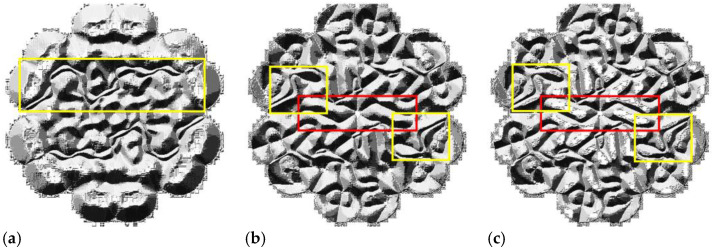
The contrast between Uniform Pattern LBP feature extraction based on the PM, the SM, and the DSM loaded with the same pistons: (**a**) the LBP feature of the PM; (**b**) the LBP feature of the SM; (**c**) the LBP feature of the DSM.

**Table 1 sensors-22-09484-t001:** The structure of ResNet-34.

Layer Name	Output Size	Residual Module	Layers
Conv1	112 × 112	0	[7×7,64], stride 2 3 × 3 max pool, stride 2
Conv2	56 × 56	3	$\left[\begin{array}{c} 3\times 3,64\\ 3\times 3,64 \end{array}\right]\times 3$
Conv3	28 × 28	4	$\left[\begin{array}{c} 3\times 3,128\\ 3\times 3,128 \end{array}\right]\times 3$
Conv4	14 × 14	6	$\left[\begin{array}{c} 3\times 3,256\\ 3\times 3,256 \end{array}\right]\times 3$
Conv5	7 × 7	3	$\left[\begin{array}{c} 3\times 3,512\\ 3\times 3,512 \end{array}\right]\times 3$
FC	1 × 1	0	average pool, FC

**Table 2 sensors-22-09484-t002:** Experimental environment.

Hardware Environment		Software Environment	
Memory	16 GB	System	Windows 11
CPU	12th Gen Intel (R) Core (TM) i7-12700H 2.30 GHz	Platform	PyCharm 2021
Graphics card	NVIDIA GeForce RTX 3060 laptop GPU	Environment	Python 3.7 (Troch main)

**Table 3 sensors-22-09484-t003:** The test results of the five sub-apertures based on different networks.

Evaluation	Model	Sub-a0	Sub-a2	Sub-a3	Sub-a4	Sub-a5	Mean
Sensing Accuracy	VGG-16	21.00%	20.10%	19.30%	20.20%	17.80%	19.68%
	Alex Net	93.90%	97.10%	88.60%	93.40%	99.30%	94.68%
	ResNet-34	98.50%	96.70%	92.10%	93.30%	88.30%	93.78%
RMSE/nm	VGG-16	38.45	35.91	37.23	34.55	36.62	36.55
	Alex Net	19.91	17.26	25.41	21.50	12.64	19.35
	ResNet-34	8.29	14.26	15.75	14.73	26.51	15.91

**Table 4 sensors-22-09484-t004:** The test results of the five sub-apertures that were based on different metrics.

Evaluation	Metric	Sub-a0	Sub-a2	Sub-a3	Sub-a4	Sub-a5	Mean
Sensing Accuracy	PM	98.50%	96.70%	92.10%	93.30%	88.30%	93.78%
	SM	95.60%	95.60%	93.50%	95.70%	99.30%	95.90%
	DSM	95.80%	98.20%	94.60%	96.40%	99.00%	96.80%
RMSE/nm	PM	8.29	14.26	15.75	14.73	26.51	15.91
	SM	13.36	7.68	13.15	10.04	7.92	10.43
	DSM	12.61	6.87	13.77	10.32	5.11	9.74

**Table 5 sensors-22-09484-t005:** The test results of the five sub-apertures that were based on the modified ResNet-34.

Evaluation	Metric	Sub-a0	Sub-a2	Sub-a3	Sub-a4	Sub-a5	Mean
Sensing Accuracy	PM	93.80%	95.80%	90.90%	94.30%	98.50%	94.66%
	SM	95.70%	98.50%	94.10%	96.90%	99.30%	96.9%
	DSM	96.30%	98.50%	95.50%	96.70%	99.30%	97.26%
RMSE/nm	PM	15.87	12.06	17.35	12.79	7.53	13.11
	SM	11.59	6.97	13.68	11.41	5.48	9.83
	DSM	10.11	6.97	12.41	9.12	6.40	9.00

## Data Availability

Not applicable.

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
