# Peer review of "Piston Sensing for Golay-6 Sparse Aperture System with Double-Defocused Sharpness Metrics via ResNet-34"

_sensors, 2022, doi:10.3390/s22239484_

Round 1
Reviewer 1 Report
In this article, a piston detection method for the sparse aperture optical system is proposed base on deep learning. It achieved attractive performances. However, further revision are necessary.
1. This article focuses on proposing a piston detection method which uses ResNet-34 as the neural network structure and Double defocused Sharpness Metric (DSM) as the input. However, authors provided two other types of input simultaneously which brings confusions. Moreover, some modifications of the network were found in Section 3. Authors are recommended to describe their best structure and method completely in Section 2 to avoid confusions. And in Section 3, comparisons between the networks or the various input types are appreciated.
2. How to calculate the sensing accuracy and the average RMSE? In Section 3.1, it was wrote “sensing accuracy is also an important criterion, the part with RMSE beyond 60 nm(0.1?) is regarded as invalid, so we can calculate the accuracy based on it.” The sensing accuracy seems named by mistake since it represents only validity. The average RMSE seems not proper. Only one group of errors were obtained corresponding to a certain sub-aperture after testing with the trained network. Error distribution can be calculated instead of RMSE distribution.
3. Authors are encouraged to give comparisons with the states of the art, or the conventional approaches further.
4. The abstract should be improved further. The distinctions and the significances of the proposed method should be emphasized on. The explanation based on Local Binary Pattern seems unnecessary.
Author Response
Please see the attachment, thanks.

Reviewer 2 Report
The authors of this paper modified the existing deep learning piston detection method for the Golay-6 array that uses a more powerful single convolutional neutral network based on ResNet-34 for feature extraction. Please address the following questions.
1. What is the sensing accuracy for this improved method?
2. What data was applied to validate the method? Please explain it in detail.
3. How long does the training based on ResNet-34 typically take?
4. What are the drawbacks of the technique presented in this paper? Please describe them in the conclusions section.
Author Response
Please see the attachment, thanks.
